# Serial grouping of 2D-image regions with object-based attention in humans

Danique Jeurissen[1]*, Matthew W Self[1], Pieter R Roelfsema[1,2,3]*

[1]Department of Vision and Cognition, Netherlands Institute for Neuroscience, Royal Netherlands Academy of Arts and Sciences, Amsterdam, The Netherlands; [2]Department of Psychiatry, Academic Medical Center, Amsterdam, The Netherlands; [3]Department of Integrative Neurophysiology, Centre for Neurogenomics and Cognitive Research, Vrije Universiteit Amsterdam, Amsterdam, The Netherlands

**Abstract** After an initial stage of local analysis within the retina and early visual pathways, the human visual system creates a structured representation of the visual scene by co-selecting image elements that are part of behaviorally relevant objects. The mechanisms underlying this perceptual organization process are only partially understood. We here investigate the time-course of perceptual grouping of two-dimensional image-regions by measuring the reaction times of human participants and report that it is associated with the gradual spread of object-based attention. Attention spreads fastest over large and homogeneous areas and is slowed down at locations that require small-scale processing. We find that the time-course of the object-based selection process is well explained by a 'growth-cone' model, which selects surface elements in an incremental, scale-dependent manner. We discuss how the visual cortical hierarchy can implement this scale-dependent spread of object-based attention, leveraging the different receptive field sizes in distinct cortical areas.

*For correspondence:
d.jeurissen@nin.knaw.nl (DJ);
p.roelfsema@nin.knaw.nl (PRR)

**Competing interests:** The authors declare that no competing interests exist.

## Introduction

Neurophysiological studies over the past 40 years have revealed that the neuronal representation of an object in low-level areas of the visual cortex consists of a set of simple features such as colors and edge orientations. However, this is not how we perceive a visual scene. Our perception is much more structured, because our visual system groups the features into objects. Introspectively, this grouping process appears to be effortless because we hardly ever perceive features in isolation. Yet, the processes for perceptual organization are only partially understood.

One influential theory suggested that perceptual grouping occurs instantaneously and in parallel if features are connected to each other, i.e. 'uniformly connected' (*Palmer and Rock, 1994*). Such a rapid grouping process would be in line with studies demonstrating that object recognition can be extremely fast and pre-attentive (*Thorpe et al., 1996*; *Moore and Egeth, 1997*; *Treisman, 1985*). High-speed object recognition presumably relies on feedforward processing, leveraging the hierarchy of features represented in the visual cortex. Neurons in lower areas coding for elementary features rapidly propagate activity to shape selective neurons in higher visual areas (*Hung et al., 2005*; *DiCarlo et al., 2012*), grouping features into more complex constellations (*Roelfsema, 2006*) so that perceptual grouping coincides with object recognition (*Biederman et al., 1987*; *Nakayama et al., 1989*; *Driver and Baylis, 1996*; *Pelli et al., 2009*). However, there also exist conditions where perceptual grouping requires a slow and serial process (*Roelfsema, 2006*).

Curve tracing is an example of a task that invokes such a serial, incremental grouping operation (*Jolicoeur et al., 1986*; *Jolicoeur and Ingleton, 1991*; *Pringle and Egeth, 1988*; *Roelfsema and*

**eLife digest** When we look at an object, we perceive it as a whole. However, this is not how the brain processes objects. Instead, cells at early stages of the visual system respond selectively to single features of the object, such as edges. Moreover, each cell responds to its target feature in only a small region of space known as its receptive field. At higher levels of the visual system, cells respond to more complex features: angles rather than edges, for example. The receptive fields of the cells are also larger. For us to see an object, the brain must therefore 'stitch' together diverse features into a unified impression.

This process is termed perceptual grouping. But how does it work? Jeurissen et al. hypothesized that this process depends on the visual system's attention spreading over a region in the image occupied by an object, and that the speed of the process will depend on the size of the receptive fields involved. If an image region is narrow, the visual system must recruit cells with small receptive fields to process the individual features. Grouping will therefore be slow. By contrast, if the object consists of large uniform areas lacking in detail, grouping should be fast. These assumptions give rise to a model called the "growth-conemodel", which makes a number of specific predictions about reaction times during perceptual grouping.

Jeurissen et al. tested the growth-cone model's predictions by measuring the speed of perceptual grouping in 160 human volunteers. These volunteers looked at an image made up of two simple shapes, and reported whether two dots fell on the same or different shapes. The results supported the growth-cone model. People were able to group large and uniform areas quickly, but were slower for narrow areas. Grouping also took more time when the distance between the dots increased. Hence, perceptual grouping of everyday objects calls on a step-by-step process that resembles solving a small maze.

The results also revealed that perceptual grouping of simple shapes relies on the spreading of visual attention over the relevant object. Furthermore, the data support the hypothesis that perceptual grouping makes use of the different sizes of receptive fields at various levels of the visual system. Further research will be needed to translate these findings to the more complex natural scenes we encounter in our daily lives.

*Houtkamp, 2011*) when participants group contour elements of an elongated curve. In the example display of *Figure 1a* subjects judge if the two colored circles fall on the same curve. In this task, reaction times increase linearly with the length of the curve. Subjects first direct their attention to the red fixation point and attention then gradually spreads over the curve until it reaches the green circle (*Scholte et al., 2001*; *Houtkamp et al., 2003*). This process is implemented in the visual cortex as the propagation of enhanced neuronal activity over the curve's representation (*Pooresmaeili and Roelfsema, 2014*) (*Figure 1b*). Curve-tracing is size invariant (*Jolicoeur and Ingleton, 1991*) so that the reaction time of observers depends little on the viewing distance. This is remarkable, because the length of the curve in degrees of visual angle increases when subjects view the stimulus from nearby. However, now the distance between the curves also increases and this enhances tracing speed (in degree/s) compensates for the longer curves so that the total reaction time remains the same. Size invariance can be explained if perceptual grouping occurs at multiple levels of the visual cortical hierarchy (*Roelfsema and Houtkamp, 2011*; *Pooresmaeili and Roelfsema, 2014*). When curves are nearby, perceptual grouping requires the high spatial resolution provided by low-level areas where neurons have small receptive fields (RFs) and horizontal connections interconnect neurons with RFs that are nearby in visual space so that progress is slow. If curves are farther apart, however, neurons in higher areas can take over and their larger RFs could speed up the grouping process (*Pooresmaeili and Roelfsema, 2014*). Size invariance also occurs when subjects solve a maze, because paths are followed at a higher speed if the distance between the walls is larger (*Crowe et al., 2000*).

Curve tracing and maze-solving might be special cases, however, and it is unclear whether perceptual grouping is serial in more typical visual scenes. We therefore investigated the time course of grouping for line drawings of relatively simple 2D shapes (*Figure 1c,d*). Specifically, we addressed

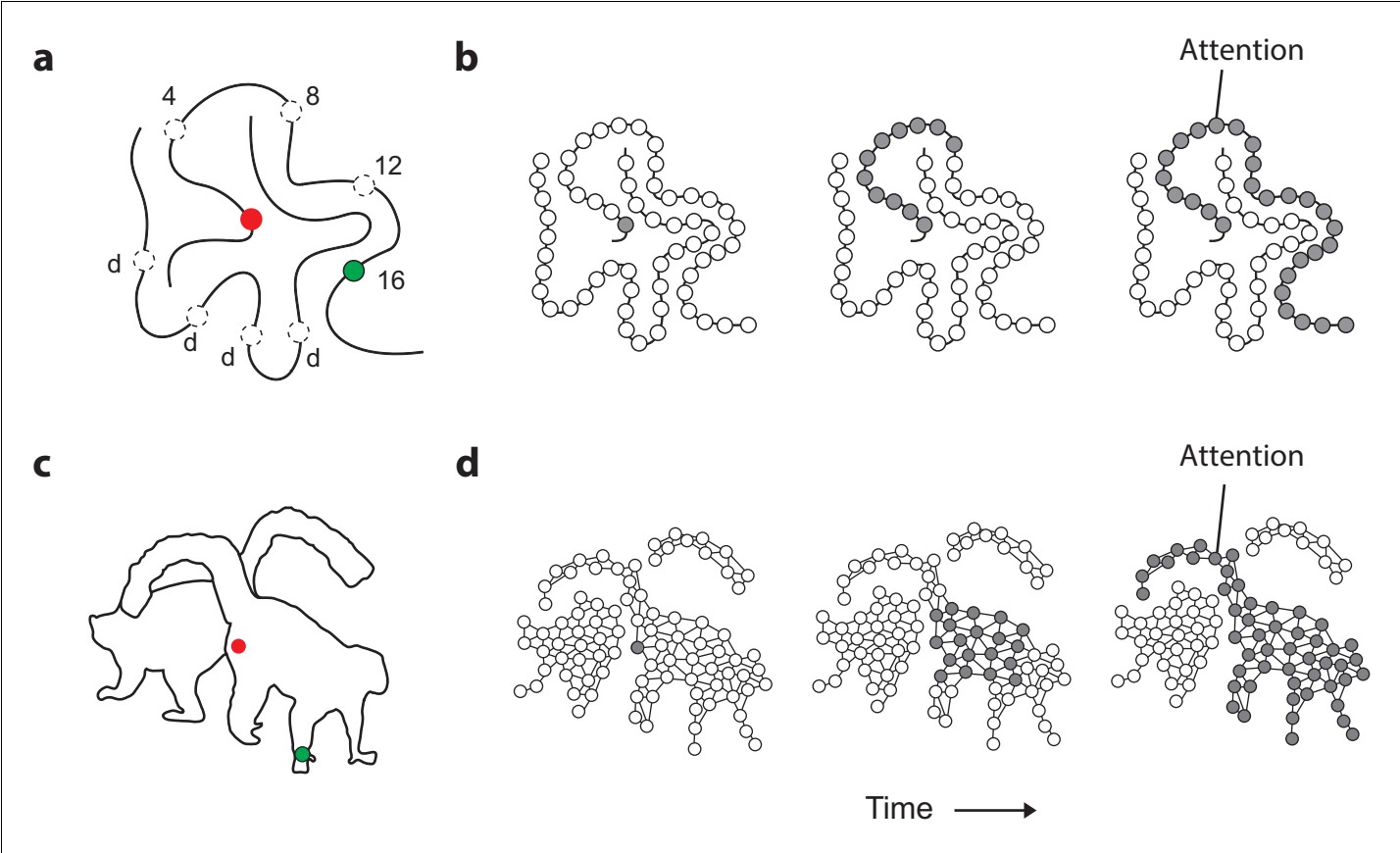

**Figure 1.** Mechanism of perceptual grouping. (a) Perceptual grouping of contour elements calls on a serial process as illustrated with a curve-tracing task. The actual stimulus contains one cue (green, dashed circles show other possible cue locations) and the participant indicates whether it falls on the same curve or on the other curve (points labeled 'd') as the fixation point. Reactions times increase linearly with the distance between the fixation point and the second cue on the same object (here 4, 8, 12, or 16 degrees) (*Jolicoeur et al., 1986*). (b) Perceptual grouping corresponds to spreading object-based attention over the curve. Cortical neurons propagate an enhanced firing rate of cells over the representation of the relevant curve in the visual cortex (*Roelfsema, 2006*). (c) An example stimulus of a 2D shape for which we measure the time course of perceptual grouping. (d) We tested the hypothesis that grouping of 2D shapes also requires a serial grouping operation.

the following questions: (1) Does perceptual grouping of simple 2D stimuli rely on a serial, incremental process? (2) What is the influence of image scale on the speed of grouping and the spread of object-based attention? (3) Is there an effect of object-recognition on the speed of the grouping process?

## Results

We examined the time-course of perceptual grouping in line drawings with a new task where subjects judged whether two cues were placed on the same object or two different objects (*Figure 2a*). Before we describe the psychophysical results, we will first outline possible models for grouping image regions, which predict different patterns of reaction times.

### Models of perceptual grouping

The uniform connectedness hypothesis by *Palmer and Rock, 1994* will serve as baseline. These authors suggested that image regions with homogenous surface properties are grouped instantaneously (*Palmer and Rock, 1994*) so that the reaction time should not depend on the placement of cues within a white region enclosed by a black contour (as in *Figure 2a*). The second model has been called 'pixel-by-pixel' in the context of curve-tracing (*McCormick and Jolicoeur, 1994*; *Jolicoeur and Ingleton, 1991*). Grouping is realized by spreading attention across pixels of the

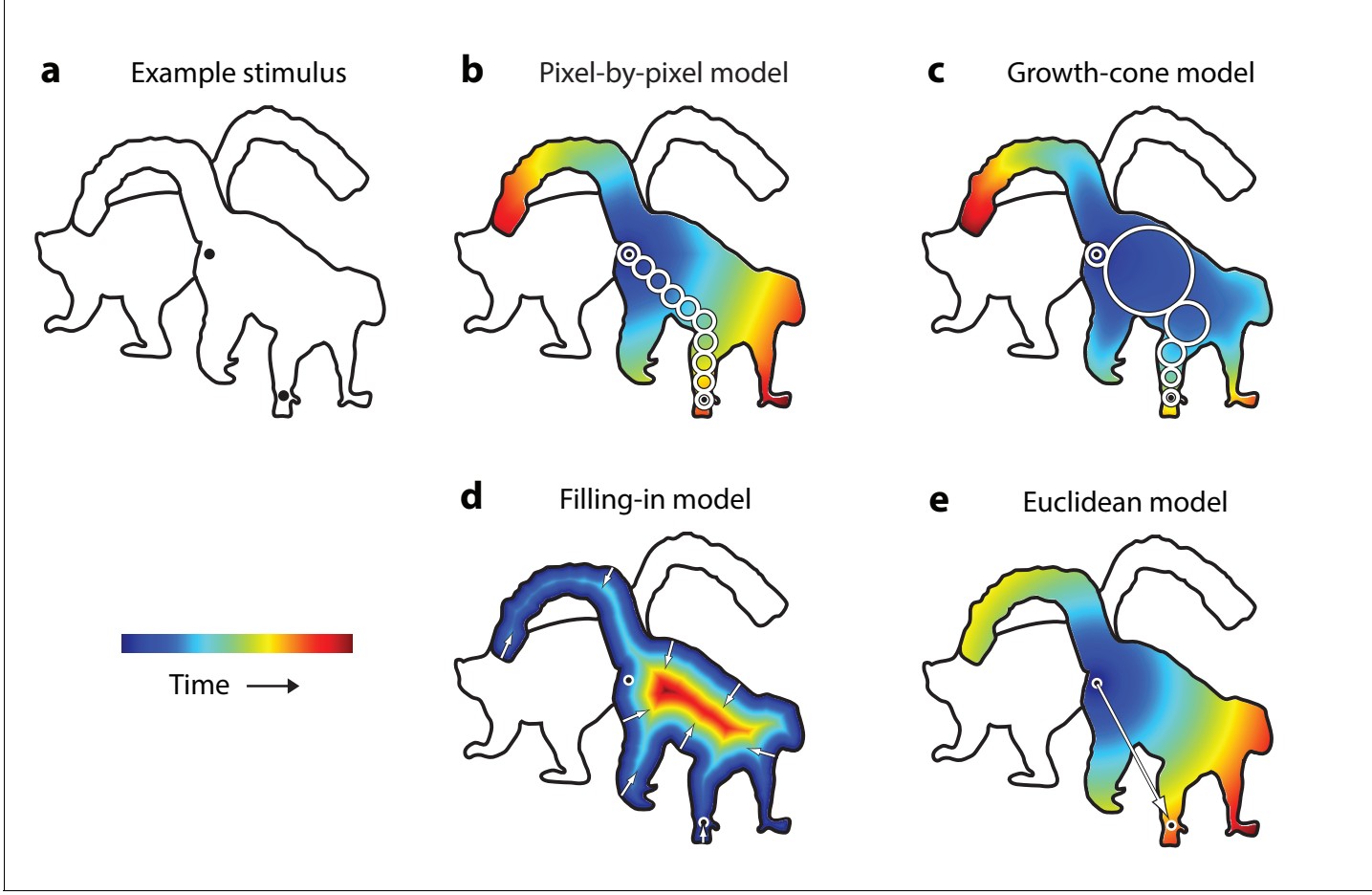

**Figure 2.** Model predictions. (a) An example stimulus. The participant indicates if the cues (black dots) fall on the same or different shapes. (b) The pixel-by-pixel model predicts that the RT depends on the shortest path within the interior of the image region. (c) The growth-cone model holds that the speed of the grouping signal also depends on the size of homogeneous image regions. (d) The filling-in model predicts that the grouping signal spreads inwards from the boundaries. (e) The Euclidean model holds that reaction time depends on the distance between cues (i.e., the eccentricity of the second cue).

same color, at a fixed speed (*Figure 2b*). Spreading starts at one of the cues and processing time is proportional to the length of the shortest path through the object. At a neuronal level, enhanced activity could spread among neurons tuned to the same color, at a single spatial scale (*Grossberg and Mingolla, 1985*). The third model is the growth-cone model (*Figure 2c*), inspired by models for curve tracing (*Pooresmaeili and Roelfsema, 2014*; *McCormick and Jolicoeur, 1994*; *Jolicoeur et al., 1986*) and visual routines (*Ullman, 1984*; *Roelfsema et al., 2000*). The growth-cone model predicts that grouping speed depends on the size of homogeneous image regions. Spreading is fast within large regions and slows down in narrow regions. In our implementation, we determined for every pixel the size of the largest circle centered at that pixel that did not touch the boundaries and assumed that speed was proportional to this size. This model is scale invariant because the total spreading time does not depend on variations in the overall size of the picture, caused e.g. by changes in viewing distance. In the visual cortex, this model could be implemented if neurons with various receptive field (RF) sizes tuned to the same feature spread the enhanced activity. In large, homogenous image regions, neurons with large RFs would make fast progress, but in narrow regions grouping speed decreases because neurons with small RFs take over. The fourth model is one where the grouping signal spreads inwards from the boundaries of the cued object so that the response time increases with the distance between the cue and the boundary (*Figure 2d*) (*Komatsu, 2006*). Such a 'filling-in' process has been proposed for brightness perception (*Paradiso and Nakayama, 1991*) and texture segregation (*Lamme et al., 1999*). The fifth and final

model is a Euclidean model, which will serve as reference because it is simple. It holds that the reaction time depends on the distance between cues, irrespective of the shape of the objects (*Figure 2e*). It differs from the pixel-by-pixel model where distance is measured as the shortest path within the object.

## Experiment 1 – Perceptual grouping of wedge-shaped objects

Experiment 1 examined the time-course of perceptual grouping with wedge-shaped stimuli (*Figure 3a*). Stimuli contained two adjacent wedges and a red fixation point that served as first cue. Twenty participants judged whether a second red cue (one of the dots shown in blue/grey in *Figure 3b* per trial) fell on the same wedge as the fixation point, by pressing a button. We measured eye-position to ensure that the participants maintained gaze on the fixation point. The screen coordinates of the cues were uninformative, because we mirrored and rotated the stimulus across trials. In Experiment 1A the second cue was at a fixed distance from the fixation point. *Figure 3c* illustrates the model predictions. The filling-in model predicts short RTs for cues near the edges and long RTs in the center. The uniform connectedness model, the Euclidean model, and the pixel-by-pixel model predict that RTs are constant, because all cues are at the same distance from fixation. In contrast, the growth-cone model predicts that RTs are short in the center and longer near the edges.

The average accuracy was 94%, without signs of a speed-accuracy trade-off (correlation between RT and accuracy; r=0.17, t(22)=0.80, p>0.4). *Figure 3d* shows the average RTs. If the two cues fell on the 'same' wedge, RTs were shortest in the center of the wedge and longer near the edges (repeated-measures ANOVA, Greenhuise-Geisser corrected, F(4.5, 85.7)=10.9, p<0.001). RTs in the 'different' condition were longer than in the 'same' condition (F(1,19)=19.5, p<0.001). A regression analysis revealed that the growth-cone model accounted for 86% of the variance of the RTs, which was significantly better than models predicting that RT is constant (p<0.001 bootstrap statistic, see Materials and methods). The Filling-in model predicted that RT is shortest near the edges, opposite to the RT data. Thus, these results support the growth-cone model and are incompatible with the other models described above.

We considered the possibility that the longer RTs near the edges were a masking effect caused by the edges. We therefore carried out a control experiment in which participants viewed the same stimulus and performed a color discrimination task. Eleven participants reported the cue's color, which could now be red or green. We did not observe a significant correlation between RT and the distance between cue and edge (r=−0.2, t(30)=−1.31, p>0.1). Thus, the edges did not act as a mask.

Experiment 1B measured the influence of the distance between the fixation point and the cue on RT in thirty new participants (*Figure 4a*). The average accuracy was 93% and we did not obtain evidence for a speed-accuracy trade off (correlation between RT and accuracy, r=0.02; t(62)=0.14, p>0.8). The average RT was higher when the cue was farther from the fixation point and a further delay was observed near the edges (warm colors in *Figure 4b*), in line with the predictions of the growth-cone model (compare *Figure 4b* to the lower left panel of *Figure 3c*). We used a regression analysis to compare models (*Figure 4c*). The filling-in model erroneously predicted that RTs are shortest near the edges. The Euclidean model (and the pixel-by-pixel model, which made the same prediction in this experiment) accounted for 49% of the variance but failed to explain the influence of edge vicinity. The growth-cone model did capture this effect and it accounted for 72% of the variance. The predictions of the growth-cone model were significantly better than those of the other models (all ps<0.01 bootstrap statistic) and the regression revealed that the average speed of perceptual grouping was 22 ms/growth cone (95% confidence interval 17–27 ms, bootstrap analysis). In other words, the RT increased by ∼22 ms for every image patch (with the size of one growth cone) that was added to the perceptual group. Thus, grouping of surfaces invokes a serial perceptual grouping process that proceeds fast in homogeneous image regions and slows down near edges, in accordance with the growth-cone model.

## Experiment 2 – Perceptual grouping of complex shapes

We next measured the time-course of perceptual grouping with more complex images to address a number of questions. First, does the shape of the object determine growth cone size? Does grouping speed decrease in narrow parts due to small growth cones and increase in broader parts?

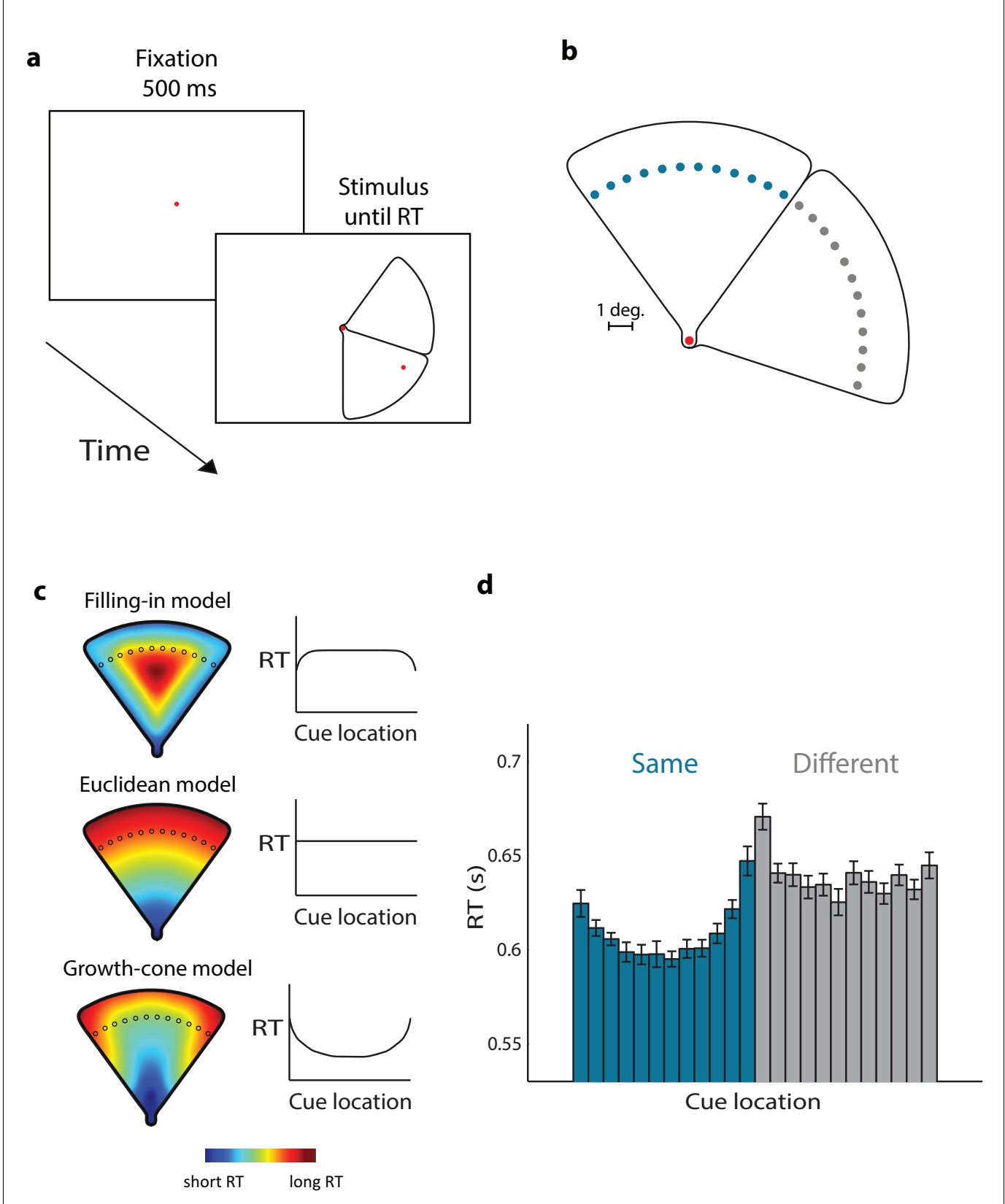

**Figure 3.** Parsing a wedge-shaped object (Experiment 1A). (a) Trial structure. After a fixation period of 500 ms, we presented two wedges and a cue. The subjects reported whether the cue fell on the same wedge as the fixation point. (b) The second cue could fall on the same (blue dots) or different

*Figure 3 continued on next page*

*Figure 3 continued*

wedge as fixation point (grey dots). The cues in the actual experiment were red. (c) RT predictions of the filling-in model, Euclidean model, and growth-cone model. (d) Average RTs of 20 observers. Blue bars, RTs for the same object; grey bars, RTs for the different object. Error bars, s.e.m. across participants (after correction for baseline differences in RT).

Second, do internal features such as textures or color changes influence grouping speed? Third, does object recognition affect perceptual grouping, because it can provide larger chunks to be grouped at once, as suggested by previous work (*Mahoney and Ullman, 1988*; *Peterson and Gibson, 1994*; *Vecera and Farah, 1997*; *Korjoukov et al., 2012*)?

We created four stimulus sets with 24 stimuli each; color pictures, detailed cartoons, cartoon outlines, and scrambled cartoons (*Figure 5—figure supplement 1*), based on 12 pictures with two animals and 12 pictures with two vehicles (*Korjoukov et al., 2012*) where animals and vehicles were easy to recognize. However, we made image recognition difficult for the scrambled cartoons by repositioning a few line segments but we left the region where the two objects intersected

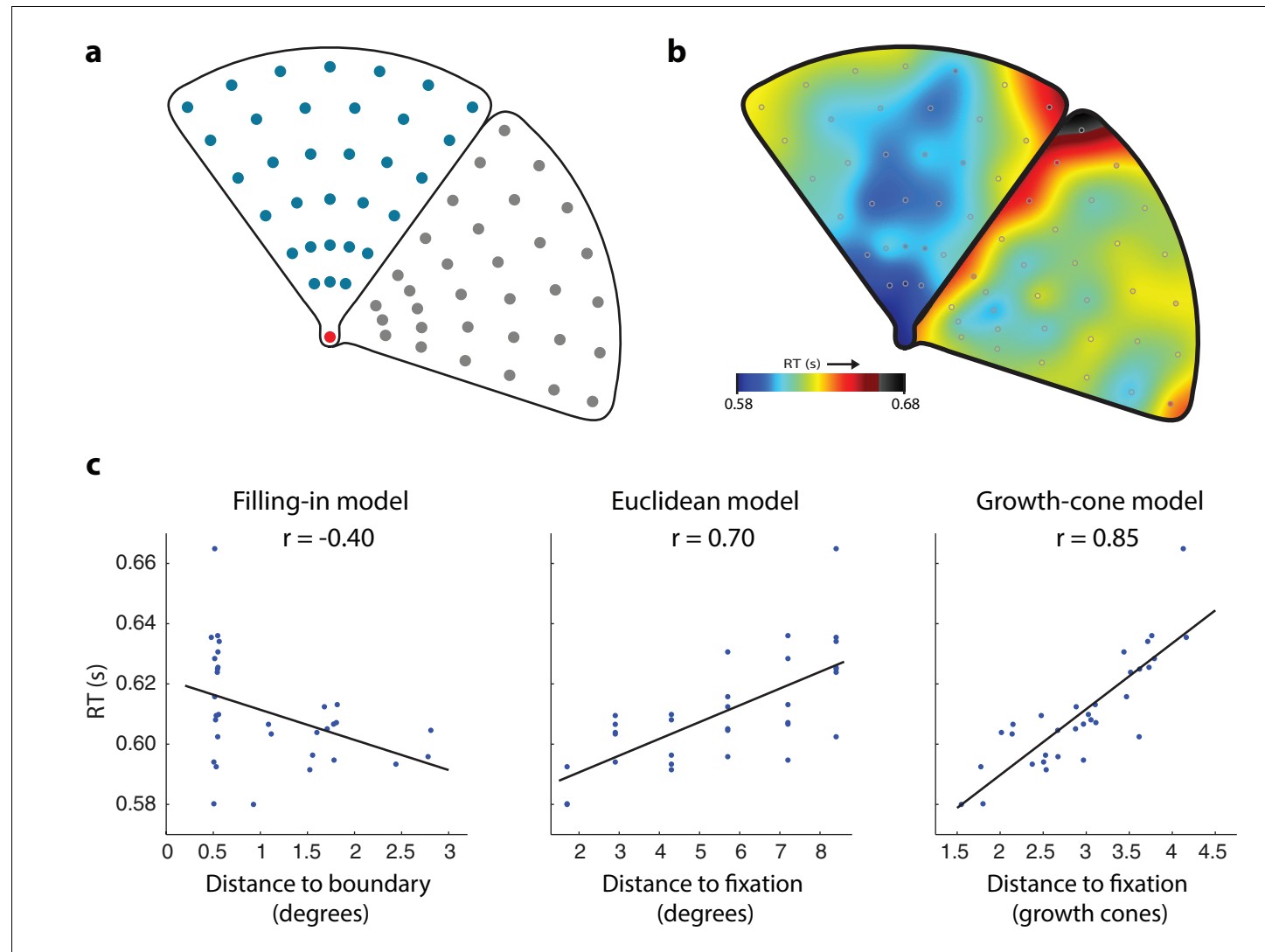

**Figure 4.** Time course of grouping a simple, wedge-shaped object (Experiment 1B). (a) In Experiment 1B cue locations were at multiple eccentricities on the same (blue) or different object (grey). (b) The pattern of RTs. Cold and warm colors show short and long RTs, respectively. (c) Regression analysis showing the fit of the models to the RT data. The growth-cone model explains 72% of the variance (r=0.85) and is superior to the other models.

unchanged (*Figure 5—figure supplement 1*, lower). Object area, perimeter, and cue locations were similar across the stimulus sets.

To measure grouping speed, we asked participants to report whether a cue appeared on the same object as the fixation point or on a different object (*Figure 5a*). When the subjects had maintained gaze on the fixation point for 500 ms, we presented the cue for 1000 ms and then the two objects. The cue was at a distance of 4.7° (1/3 of the trials) or 9.4° (2/3 of the trials) from the fixation point and these distances were matched if cues fell on the other object. With 24 images per condition and 3 cue positions per object, we had 72 data points for the same object condition, and an equal number for the different object condition. We assigned a total of 88 participants to the four image categories (between-subject condition), with 22 participants per condition. Sixty-four

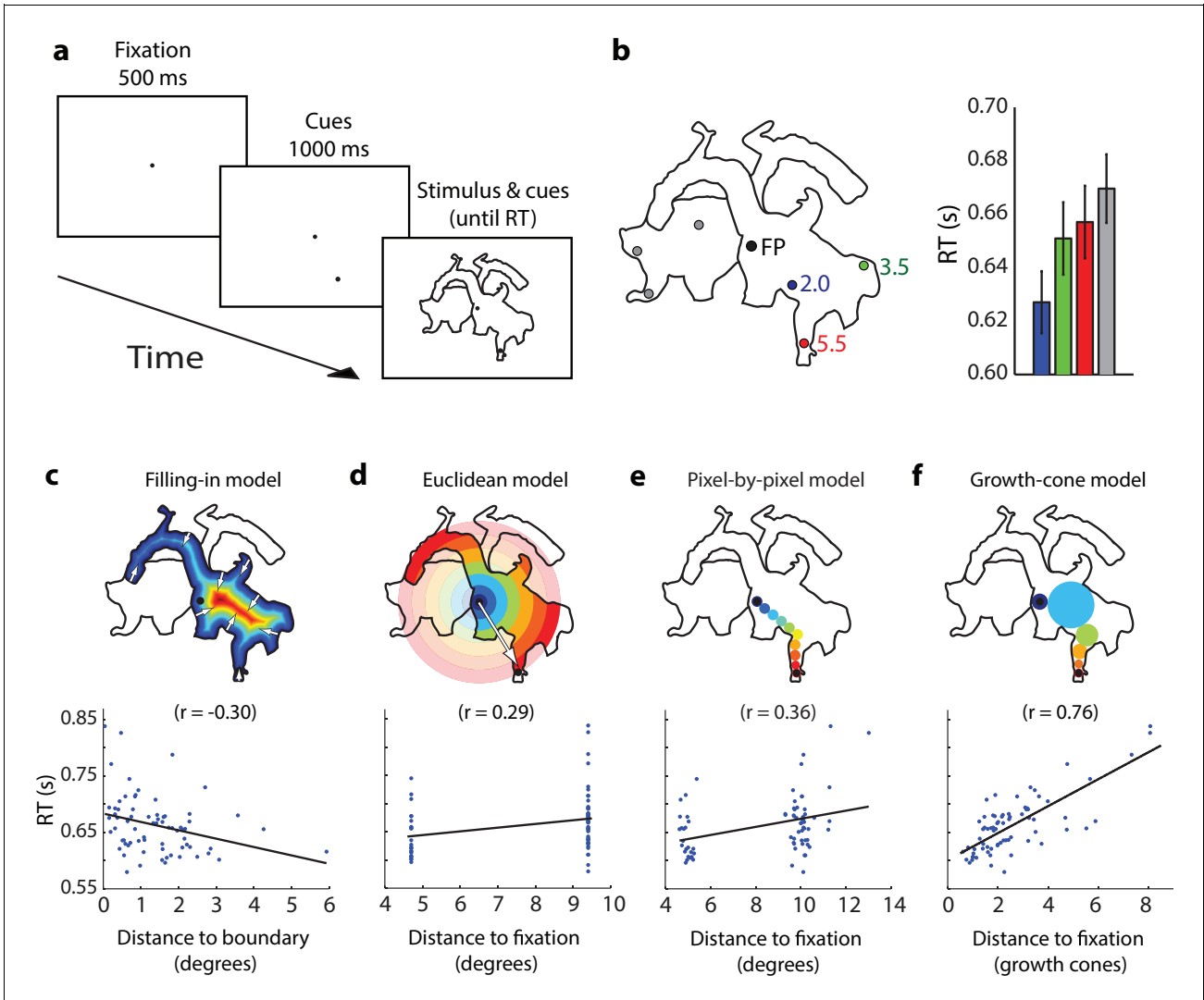

**Figure 5.** Time course of parsing scrambled cartoons (Experiment 2) (the actual stimuli were white outlines on a black background). (a) After a fixation epoch of 500 ms, the subjects saw the cue for 1000 ms, and then also two objects. They reported whether the cue fell on the same object as the fixation point. (b) Left, Example stimulus. FP, fixation point. Numbers indicate the estimated number of growth cones between the cues and the FP. Right, RTs averaged across participants for the different cue locations. Error bars represent s.e.m. (c-f) Regression of the RT on the predictions of the filling-in (c), Euclidean (d), pixel-by-pixel (e), and growth-cone model (f).

The following figure supplement is available for figure 5:

**Figure supplement 1.** The full stimulus set for Experiment 2.

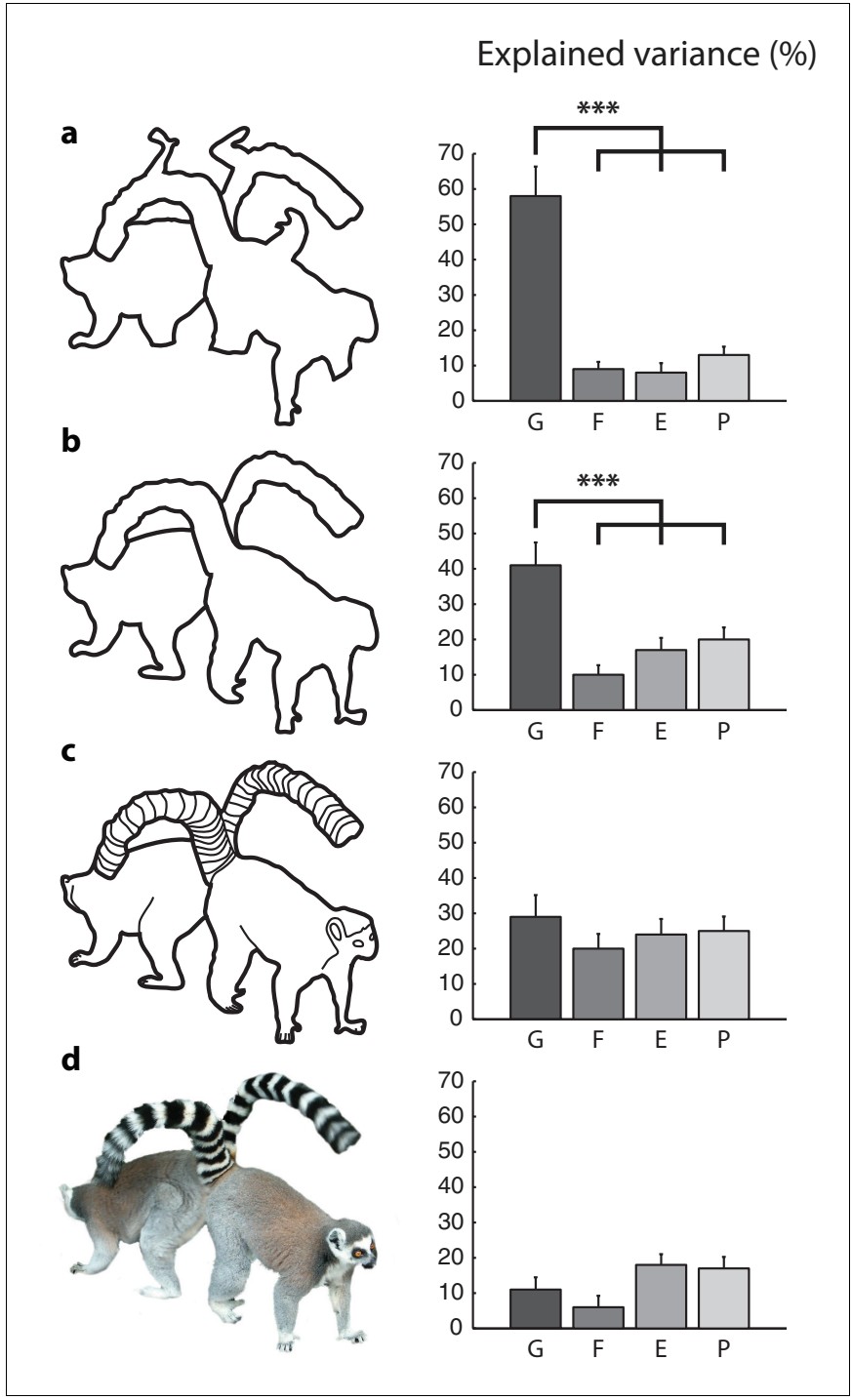

**Figure 6.** Model fits for the different types of images (Experiment 2). The bars show the variance in RT explained by the different models for (**a**) scrambled cartoons; (**b**) cartoon outlines; (**c**) detailed cartoons; and (**d**) color pictures. G, growth-cone; F, filling-in; E, Euclidean; P, pixel-by-pixel model. Asterisks (***) represent $p < 0.01$ (bootstrap test).

participants maintained gaze on the fixation point until the response, but we found that eye-movements had little influence on the pattern of RTs (Materials and methods).

*Figure 5b* shows an example scrambled cartoon picture and the RTs averaged across 22 participants. The RT was shortest for the nearest cue (blue in *Figure 5b*, 627 ms) and it increased by 24/

30 ms (green/red in *Figure 5b*) for the more distant cues. We investigated how well the different grouping models account for the pattern of RTs across all scrambled cartoon pictures with a regression analysis (*Figure 5c–f*). As in experiment 1, the filling-in model made the wrong prediction that RT is shortest near boundaries (r=−0.30). The Euclidean model predicts that RT depends on the distance between fixation point and cue (i.e. the eccentricity of the cue) but this relationship was weak in the data (r=0.29). The predictions of the pixel-by-pixel model were better (r=0.36) but those of the growth-cone model were best (r=0.76). The growth-cone model explained 58% of the variance, which was significantly better than the other three models (*Figure 6a*) (bootstrap statistic: all ps<0.01) and the regression yielded an estimated time per growth cone shift of 24 ms (95% confidence interval; 18–29 ms), in accordance with the estimate of Experiment 1. Thus, perceptual grouping of 2D image regions invokes a serial grouping process with a speed that depends on their scale: wide regions are grouped quickly and narrow regions more slowly.

We constructed the scrambled cartoons with the aim to make the objects difficult to recognize, as object-recognition can influence perceptual grouping. We did, however, also test the cartoon images in 22 subjects (*Figure 6b*). The growth cone model explained 41% of the variance, which is more than any other model (bootstrap: all ps<0.01). However, the growth cone model was not better than the other models for the detailed cartoons with interior contours and for the color pictures (*Figure 6c,d*).

These results reveal that the growth-cone model provides an excellent account for the pattern of RTs for scrambled cartoons and cartoon outlines, but that the predictions for the detailed cartoons (with internal contours) and colored pictures are worse. It follows that the interior lines of the detailed cartoons and the colors and textures of the picture stimuli influence grouping speed. We only used the outlines of the shapes to estimate growth-cone size, which may explain why predictions worsened. Future grouping models might explain additional variance by taking interior features, texture gradients and color transitions into account, but these new models would require additional experimental and theoretical work.

## Experiment 3 – The role of object-based attention in perceptual grouping

Previous studies suggested that grouping depends on the spread of object-based attention across the features that need to be bound in perception (*Rensink, 2000*; *Roelfsema, 2006*; *Roelfsema et al., 2007*). So far, our experiments measured reaction times but we did not test the spread of 'attention'. Our third experiment measured the spread of object-based attention, capitalizing on the Egly cueing paradigm (*Egly et al., 1994*) to measure the speed. *Egly et al., 1994* used displays similar to the ones in *Figure 7a*. Their participants saw two bars, then one bar was cued at one of its ends and finally a target was presented to which the participants responded with a button press. If the cue was valid because the target appeared at the same location as the cue, the participants' responses were faster than if it appeared at another location, because spatial attention was summoned by the cue. Their main finding, however, was that if the cue was invalid and the target appeared at the other end of the cued bar (invalid same-object trials), the response was faster than if it appeared at the non-cued bar (invalid different-object trials), even if the distance between the target and the invalid cue was constant. The reaction-time difference between invalid same- and different object trials is a measure for object-based attention, which is hypothesized to select the entire cued bar (*Egly et al., 1994*; *Xu and Chun, 2007*; *Lamy and Egeth, 2002*; *Drummond & Shomstein 2010*).

In our version of the experiment, eleven participants maintained gaze at a fixation point (controlled with an eye-tracker) and they saw two horizontal or vertical bars. We cued a corner of one of the bars (*Figure 7a*) and in 80% of trials the participants reported the appearance of a red target dot at one of three positions by pressing a button. The remaining 20% were catch trials that did not require a button press. In target trials, we presented the target dot at the cued location (60%, valid trials), at the opposite side of the same bar (20%, invalid same object), or in the other bar (20%, invalid different object). We varied the time between cue and target onset (between 200 and 600 ms) to measure the time-course of the object-based advantage (invalid same trials vs. invalid different trials). To examine if the speed of object-based attention depends on the size of image regions, we presented either broad bars (3.9° wide, red panels in *Figure 7a*) or narrow bars (2°, blue panels). The growth-cone model predicts that the spread of object-based attention is faster for

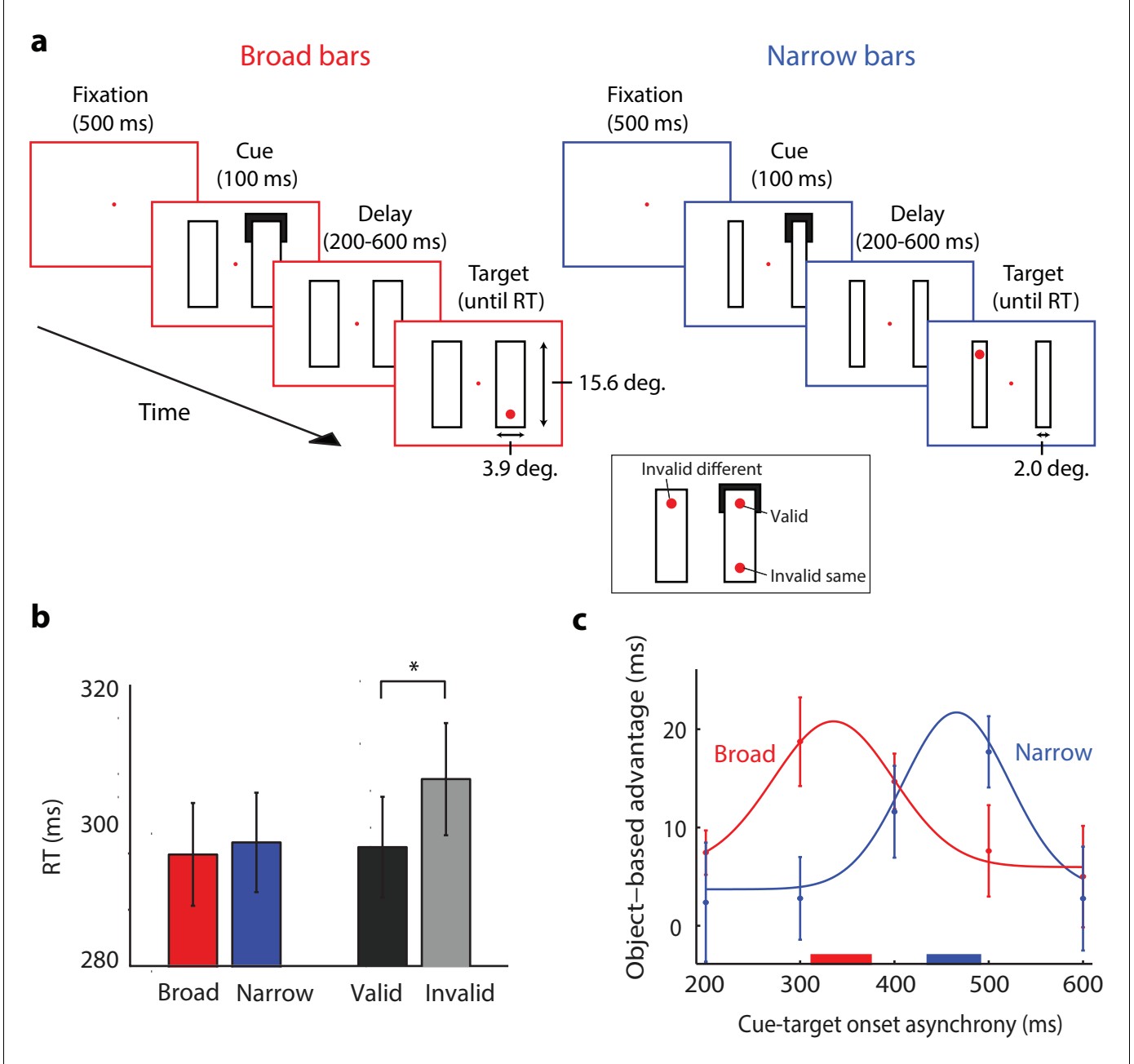

**Figure 7.** The time-course of spreading object-based attention (Experiment 3; actual stimuli were white on a black background). (**a**) We presented two broad (left) or narrow bars (right) and one bar was cued at one of its ends for 100 ms. After a variable delay, we presented a target dot at the cued location (valid trials, see inset) or at one of two locations that were equidistant from the cue, on the same (invalid same) or on the other bar (invalid different), i.e. only one target dot per trial. In catch trials (not shown), the target did not appear. (**b**) Reaction times for the validly cued trials for the broad (red) and narrow bars (blue) did not differ (t-test, p>0.1). Reaction times for the validly (black) cued locations were faster than those for the invalidly (grey) cued locations (*, p<0.01 in t-test). (**c**) The object-based advantage ($RT_{invalid\ different} - RT_{invalid\ same}$) as a function of cue-target onset asynchrony. The curves show the fit of Gaussian functions to the object-based advantage for the broad (red) and narrow bars (blue), respectively. The red and blue horizontal bars on the x-axis indicate the 95% confidence interval of the peak of the Gaussian function as measured with a bootstrap method. Error bars represent s.e.m. of the data points.

broader bars because growth-cones are larger; we estimated that there were 3.3 and 7.2 growth-cones between cue and target for the broad and narrow bar, respectively (see Materials and methods).

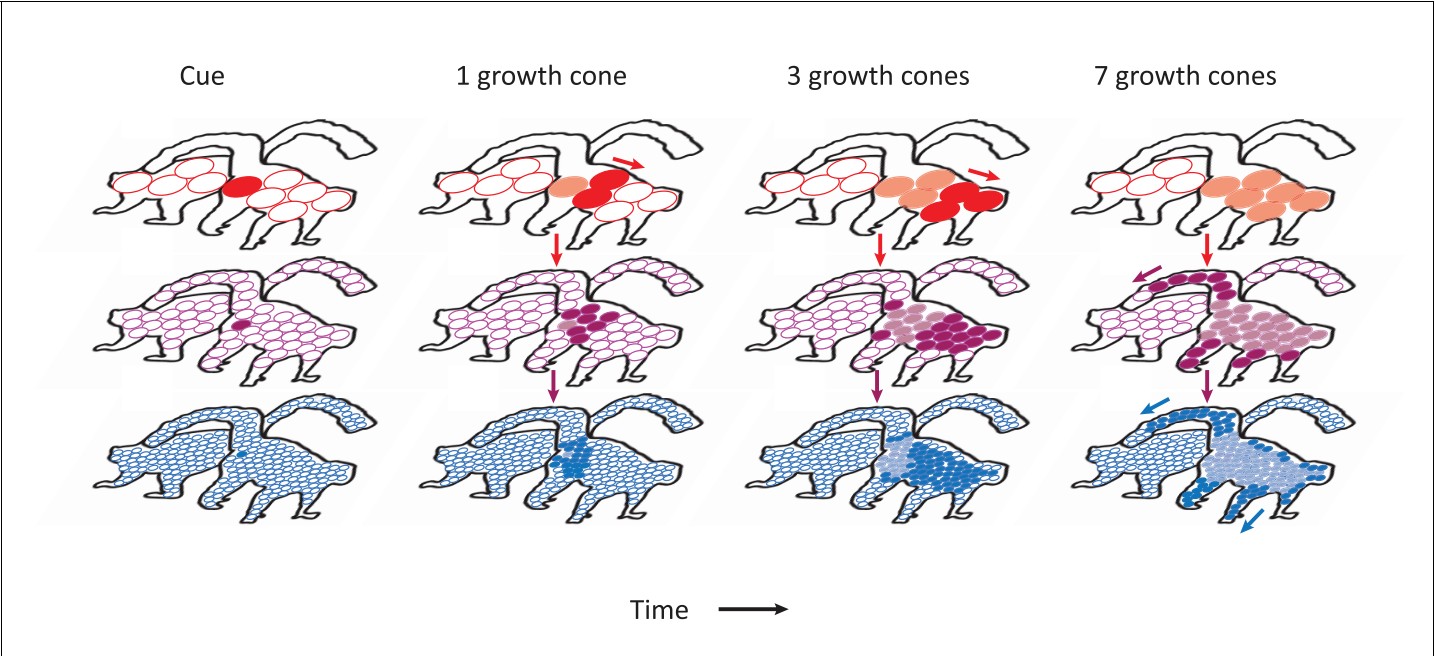

**Figure 8.** Schematic representation of the contribution of different visual areas to perceptual grouping. Horizontal rows illustrate low-, mid-, and high-level visual areas with larger receptive fields in higher areas. The four columns illustrate different time steps during the grouping process; after the presentation of the cue, and after 1, 3 and 7 growth-cone shifts. The labeling process begins at the cued location. Higher cortical areas with large receptive fields make great strides in the propagation of enhanced neuronal activity and this fast progress also impacts on lower areas through feedback connections (downward pointing arrows). However, the higher visual areas cannot resolve fine-scale details and the grouping of narrower image regions therefore relies on the propagation of enhanced neuronal activity in lower visual areas with smaller receptive fields. Darker colors represent image regions that have been recently reached by the grouping process and lighter colors denote image regions that were labeled at an earlier point in time. White circles represent receptive fields that have not been reached by the grouping process. Note that the labeling process is serial and that the speed of grouping depends on the size of the receptive fields that contribute to the grouping process.

The average accuracy across participants was 99%. On validly cued trials, bar width did not influence RT ($t(10)=-1.450$, $p>0.1$; *Figure 7b*). As expected, RTs were shorter on validly cued than invalidly cued trials ($t(10)=-3.468$, $p<0.01$). We computed the object-based advantage for the broad and narrow bars and found that RTs were shorter with invalid cues on the same object than on a different object, an object-based advantage that is in accordance with previous studies (*Egly et al., 1994*; *Xu and Chun, 2007*; *Lamy and Egeth, 2002*; *Drummond & Shomstein 2010*). The maximal object-based advantage occurred earlier for the broad bar than for the narrow bar (*Figure 7c*). We fitted a Gaussian function to the object-based benefit and found that it peaked after 344 ms (95% confidence interval: 311–376 ms) for the broad bar and after 463 ms (434–492 ms) for the narrow bar ($p<0.001$; bootstrap analysis). We estimated the speed of attention spreading by dividing the time difference of 119 ms by the difference in the estimated number of growth cones (7.2–3.3=3.9) and obtained a speed of 31 ms per growth cone, compatible with the speed estimates of Experiments 1 and 2.

## Discussion

The new task allowed us to measure the time course of grouping operations in simple 2D line drawings and revealed that perceptual grouping relies on a serial, incremental process, refuting the uniform connectedness rule proposed by *Palmer and Rock, 1994*. A growth-cone model where the speed of the grouping operation depends on the size of homogeneous image regions best describes the results. At first sight, such a slow, serial grouping process seems to be at odds with previous findings that some forms of perceptual grouping, related to the recognition of objects or image categories like animals and vehicles, is very rapid and pre-attentive (*Thorpe et al., 1996*; *Kirchner and Thorpe, 2006*; *Moore and Egeth, 1997*; *Treisman, 1985*). The pre-attentive

formation of perceptual groupings appears to rely on rapid feedforward computations that take place when visual information propagates from early to higher visual areas (*Yamins et al., 2014*), a strategy that is also used by state-of-the-art convolutional networks for object recognition (*Szegedy et al., 2015*). However, the recognition of objects in the scene does not imply that visual segmentation processes have completed (*Epshtein et al., 2008*; *Houtkamp and Roelfsema, 2010*). For example, connectedness is one of the cues for perceptual grouping, and it is difficult to compute connectedness in a purely feedforward manner (*Minsky and Papert, 1990*; *Roelfsema et al., 1999*). Furthermore, if a scene contains multiple animals, the recognition of the species does not imply that all parts of every animal have been grouped as coherent percepts (*Korjoukov et al., 2012*) because there are also cases where recognition precedes image parsing (*Vecera and Farah, 1997*; *Peterson et al., 1991*). Experiments 1 and 2 specifically probed this grouping process, asking participants to report whether cues fell on parts of the same object, and we obtained unequivocal evidence for a serial grouping operation. Our results suggest that the visual system processes simple 2D image regions as miniature mazes, and for mazes serial processing is less surprising (*Crowe et al., 2000*).

Serial processing has also been observed previously in contour grouping tasks (*Roelfsema, 2006*), where grouping is associated with the gradual spread of object-based attention over the perceptual group (*Figure 1a,b*) (*Houtkamp et al., 2003*), a process that can only occur for one curve at a time (*Houtkamp and Roelfsema, 2010*). In the visual cortex, the spread of attention is associated with the propagation of enhanced neuronal activity over the representation of the relevant curve (*Pooresmaeili and Roelfsema, 2014*). Spreading is fast if curves are far apart but it slows down if curves are in each other's vicinity. The curve-tracing operation has also been modeled as a growth-cone process, similar to the one we now used to model grouping of 2D image regions (*Pooresmaeili and Roelfsema, 2014*). In daily life, this serial grouping operation may only be necessary for specific tasks. During object recognition, image parsing may be unnecessary, but during grasping one needs to determine whether surfaces belong to the same object to avoid placing fingers on surfaces of different ones.

## A growth-cone model for parsing 2D-image regions

The new growth-cone model holds that grouping speed depends on the size of homogenous image regions. The variation in the scale of processing could rely on multiple stages of the visual cortical hierarchy where neurons have different RF sizes (*Figure 8*). Near boundaries and in narrow parts of an object, the perceptual grouping process requires a high spatial resolution, which can be supported by small receptive fields in lower areas (purple and blue in *Figure 8*). In these areas the horizontal connections, responsible for the spread of enhanced activity, link neurons that represent nearby locations in visual space. If the homogeneous image regions are larger, the propagation may take place in higher areas with larger RFs so that the grouping speed is higher (red in *Figure 8*). This scale dependence of the spread of object-based attention is in accordance with neuroimaging studies revealing the stronger engagement of lower visual areas when the relevant spatial scale is finer (*Rijpkema et al., 2008*; *Hopf et al., 2006*). Future studies could aim to unravel the interactions between neurons at different levels of the visual cortical hierarchy during image parsing.

Although the growth-cone model capitalizes on the presence of receptive fields with different sizes at distinct levels of the visual cortical processing hierarchy, it does not require the receptive field size to be constant within a visual cortical area. Indeed, it is well known that the size of receptive fields increases with eccentricity. Suppose that the task is to group an elongated bar with a width of 2° (as in *Figure 7a*). The propagation of enhanced neuronal activity would take place in area V1 for the more eccentric parts of the bar (e.g. at 10° eccentricity) (*Gattass et al., 1981*) but area V4 would make fastest progress where the bar is nearer to the fovea (e.g. at 2° eccentricity) (*Gattass et al., 1988*). The horizontal distance between neurons with abutting, non-overlapping RFs is relatively constant across visual areas (*Harvey and Dumoulin, 2011*). Thus, if we assume that the propagation of enhanced neuronal activity through horizontal connections is also similar, it follows that the propagation speed (in °/s or growth-cones/s) should also be constant across eccentricities for a bar with a constant width. The growth-cone mechanism is therefore compatible with the fact that the receptive field size in visual cortical areas increases with eccentricity.

## Influence of object recognition and interior contours on image parsing

The current implementation of the growth-cone model is based on the outlines of the shapes only and, accordingly, the interior features of detailed cartoons and color pictures decreased fit quality. Consider, for example, the picture with lemurs of *Figure 6c*. The interior contours in the tails form closed, disconnected compartments that cannot be grouped by the growth-cone model. The overall influence of the interior contours is difficult to predict, however. On the one hand, they might slow down parsing by presenting barriers. On the other hand, they might facilitate object recognition and thereby speed up the grouping. Hence, the interior contours, the influence of color and texture, as well as the influence of object-recognition go beyond the simple models considered by us so that variance had to remain unexplained.

Importantly, perceptual grouping with the same color pictures also relies on a serial process (*Korjoukov et al., 2012*). It would therefore be of interest to generalize the growth-cone model to account for perceptual grouping of objects with interior contours, colors, and textures. We expect that these generalizations necessitate a mechanism for object recognition. In the picture of the lemur (*Figure 6c,d*), for example, we see that the left tail belongs to the right lemur, because we know what a lemur looks like. Segmentation based on the characteristic shape of objects is known as 'semantic segmentation'. Models of segmentation can take advantage of the recent breakthroughs in hierarchical convolutional neural networks for object recognition (e.g. *Krizhevsky et al. (2012)*; *Zeiler and Fergus (2013)*; reviewed by *LeCun et al., 2015*). These networks consist of multiple hierarchically organized layers, which transform simple features at the lower layers, resembling lower visual cortical areas, to representations for shape recognition in higher layers, resembling the inferotemporal cortex (*Güçlü and van Gerven, 2015*; *Yamins and DiCarlo, 2016*). Recent studies have started to apply hierarchical convolutional networks to achieve semantic segmentation (*Hong et al., 2015*; *Noh et al., 2015*). The resulting networks can segment pictures of everyday visual scenes. They first use a convolutional network to determine the semantic categories present in the visual scene, which is followed by a 'deconvolutional network' that retrieves the lower-level features that are part of a particular semantic category. Hence, these models explain why object-recognition facilitates image parsing in human perception (*Vecera and Farah, 1997*; *Peterson et al., 1991*; *Korjoukov et al., 2012*). The deconvolutional network could be implemented in the visual cortex as a top-down influence from neurons in shape selective cortical areas to neurons in lower visual areas (i.e. generalizing the feedback connections in *Figure 8*) (*Hochstein and Ahissar, 2002*). We expect that semantic segmentation networks may need to also include horizontal connections in order to account for the serial patterns of reaction times in grouping tasks with natural images (as in *Korjoukov et al., 2012*). According to this view, the parsing of a natural scene would invoke an incremental grouping process that gradually adds simple features and more complex shape fragments to an evolving perceptual group (*Roelfsema, 2006*; *Roelfsema and Houtkamp, 2011*).

## The spread of object-based attention

We conjectured that grouping relies on the spread of object-based attention within homogenous image regions, and this spread should therefore depend on the size of these regions. Our third experiment confirmed this prediction by showing that the spread of object-based attention is slower for narrow regions. The last experiment yielded an estimate of the attentional propagation speed of ~30 ms per growth cone, which is similar to the estimates of 22 and 24 ms/growth cone of the other two experiments. In the cortex, a shift time of ~25 ms would correspond to the propagation of activity between neurons with abutting, non-overlapping RFs. In V1, the distance between neurons with adjacent RFs is 2–4 mm, and it is only slightly larger in higher visual areas (*Hubel and Wiesel, 1974*; *Harvey and Dumoulin, 2011*). Our results therefore suggest that the propagation speed is 8–16 cm/s (2–4 mm/ 25 ms), which falls neatly in the range of previous neurophysiological measures for the horizontal propagation of neuronal activity in visual cortex (*Hirsch and Gilbert, 1991*; *Nauhaus et al., 2009*; *Sato and Carandini, 2012*).

## Conclusion

Our results suggest that perceptual grouping of 2D image regions calls on a serial process that takes tens to hundreds of milliseconds for naturalistic pictures. Subjects apparently need this time to spread object-based attention within in the interior of image regions that need to be grouped in

perception. The typical time between two saccadic eye movements is $\sim$200–300 ms and it is therefore tempting to speculate that the duration of visual fixations is optimized to allot time for both object recognition and image parsing before the next saccade is made.

## Materials and methods

### Model implementation

#### Euclidean model

The Euclidean model holds that RT depends linearly on the distance between cues, $p_1$ and $p_2$:

$$RT_{Eucl} = RT_{Offset} + C \cdot \| p_1 - p_2 \|, \tag{1}$$

where $RT_{Offset}$ is the intercept and $C$ the slope (in ms/deg).

#### Filling-in model

The filling-in model assumes that the grouping signal spreads inwards from the edges of the object towards the interior. RT therefore depends on the distance of the cues to the object boundary:

$$RT_{Fill} = RT_{Offset} + C \cdot \max \left( \| p_1 - p_{edge1} \|, \| p_2 - p_{edge2} \| \right) \tag{2}$$

where $RT_{Fill}$ is the predicted reaction time, $RT_{Offset}$ is the intercept and $C$ the slope (in ms/deg), and $p_{edge1/2}$ is the point on the object's edge closest to the respective cue. Thus, $RT_{Fill}$ depends on the largest cue-edge distance because the grouping signal has to reach both cues.

#### Pixel-by-pixel model

In the pixel-by-pixel model the object is grouped incrementally by spreading object-based attention at a constant speed over the object's surface. The RT depends on the distance between cues, but now measured within the object's interior. To compute the shortest path between the cues, we defined an 8-connected lattice $G=(V,E)$ with vertices (pixels) $V$ connected by edges $E$ where two neighboring pixels are only connected if they do not cross the object boundary. A path is a set of connected edges $\{e \epsilon E\}$. We defined P as the set of all paths that emanate from $p_1 \epsilon V$ and terminate at $p_2 \epsilon V$. An edge has length 1 for two horizontally and two vertically aligned pixels and $\sqrt{2}$ for diagonally aligned pixels. RT was estimated as

$$RT_{Pix} = RT_{Offset} + C \cdot \min_{p \in P} \sum_{e \in p} f_{Pix}(e), \tag{3}$$

where $f_{Pix}(e) = 1$ for horizontally and vertically aligned pixels and $f_{Pix}(e) = \sqrt{2}$ for diagonally aligned pixels. The pixel values are transformed into visual degrees. The shortest path in P was computed with Dijkstra's algorithm (*Dijkstra, 1959*).

#### Growth-cone model

The growth-cone model is similar to the pixel-by-pixel model, but it holds that grouping occurs at multiple spatial scales, in visual areas with different RF sizes. The model selects a scale for the growth cone (i.e. RF size) at every pixel on the surface, which corresponds to the diameter $d_{inscr}$ of the largest inscribed circle centered on that pixel which fits within the boundaries of the object. The growth-cone size determines the speed of the spreading processes at that location; larger growth cones correspond to higher speeds. Accordingly, RT is estimated as follows:

$$RT_{GC} = RT_{Offset} + C \cdot p \in P \min \sum_{e \in p} f_{GC}(e), \tag{4}$$

where $RT_{Offset}$ is the intercept and $C$ the slope (in ms/growth-cone) of the regression, $f_{GC}(e) = \frac{1}{(d_{inscr}(pix_1) + d_{inscr}(pix_2))/2}$ for horizontally and vertically aligned pixels and $f_{GC}(e) = \frac{\sqrt{2}}{(d_{inscr}(pix_1) + d_{inscr}(pix_2))/2}$ for pixels aligned diagonally and $RT_{GC}$ is the reaction time. $d_{inscr}(pix_1)$ and $d_{inscr}(pix_2)$ are the maximal diameters of inscribed circles centered at two adjacent pixels. We used a minimum $d_{inscr}$ of 1 degree, roughly corresponding to the smallest RF sizes in V1. $RT_{GC}$ is scale invariant. If the image is

magnified by a factor, the distance between two cues increases, but the size of the growth cones increases by the same proportion such that $RT_{GC}$ stays the same.

## Experiments

The participants reported normal or corrected-to-normal acuity and were paid for their participation. The Ethics Committee at the University of Amsterdam approved the experiments. Informed consent was obtained before the start of the experiment. Eye position was sampled at 1000 Hz with an Eye-Link eye-tracking system (SR Research Ltd). The experiments were performed in a dimly lit room. We used a chinrest and the participants sat at a distance of 57 cm from a CRT screen. They practiced the task before data collection started.

## Experiment 1

### Participants

Twenty participants enrolled in Experiment 1A and 30 participants in Experiment 1B. Their average age was 24 years (range: 18–55; 35 females, 48 right-handed).

### Stimuli and procedure

The stimulus consisted of two wedge shapes, drawn in black on a white background (*Figure 3a*). When the subject had maintained gaze at the red fixation (size, 0.3°) point for 500 ms, the wedges and the second red cue appeared. The two wedges were always adjacent and the participant's task was to indicate whether the cues were on the same or different wedges by pressing a button. The stimulus was presented until a response was given or maximally for 3000 ms. We gave feedback ('correct', 'incorrect', or 'too late' – for responses after 3000 ms) after each trial. We instructed participants to complete the task as accurately and quickly as possible and we repeated incorrect trials later in the block.

The two wedges only differed near the fixation point, at the narrowest part of the wedge (size, 0.6°). The wedges had a radius of 9° (measured from the fixation point) and a width of 10°. In Experiment 1A, the second cue (0.3°) could be placed at one of 24 equidistant positions on the same or different wedge (50% same and 50% different), at a distance of 7.2° from fixation (*Figure 3a,b*). The minimal distance between cue and wedge outline was 0.3°. The wedges appeared at one of 9 rotation angles around the fixation point and the wedge with the fixation point was adjacent to the other wedge in the clockwise or counterclockwise direction. We presented these conditions (9 rotations, 2 mirror images, 24 cue locations) in a pseudo-randomized order with 4 repetitions per condition, resulting in a total of 1728 trials per participant, presented in 12 blocks of 144 trials each.

In Experiment 1B, the second cue could be placed in 64 positions. Within each wedge, three of these locations had an eccentricity of 1.7° from fixation, five locations of 2.9°, five of 4.3°, six of 5.7°, six of 7.2°, and seven of 8.4°. Cue locations at each eccentricity were equidistant from each other and at a minimal distance of 0.5° from the wedge outline (*Figure 4a*). The wedges appeared in one of 8 rotations. We presented the conditions (8 orientations, 2 mirror images, 64 cue locations, 2 repetitions) in a pseudo-randomized order with 2048 trials per participant, in 16 blocks of 128 trials.

Whenever fixation was lost (gaze deviated from fixation by more than 1.4 degrees), the trial was aborted and repeated later within the same block. The buttons (left and right) for the response (same or different) were counterbalanced across participants.

### Data analysis

We analyzed RTs of correct trials after removing outliers separately for each cue condition if they deviated from the mean of the 1/RT distribution by more than 2.5 s.d. (<0.5% of the data). We tested for a possible speed-accuracy trade-off by computing the correlation between the average RT and accuracy across cue conditions. We used a repeated-measures ANOVA to test RT predictions in Experiment 1A. We applied the Greenhouse-Geisser correction if appropriate.

For Experiments 1A and 1B, we performed a regression analysis to test how well $RT_{Fill}$, $RT_{Eucl}$/$RT_{Pix}$, and $RT_{GC}$ (defined in equations above) predicted RT. All models have two free parameters, the baseline reaction time $RT_{Offset}$, and the slope $C$ and we used a bootstrapping procedure to test whether the quality of the fits differed between the models. We resampled the data 10,000 times by drawing a random sample of 30 participants with replacement, maintaining the within-subject cue-

condition assignment. We computed the distribution of differences in explained variance between models (R-squared) and the confidence interval of the slope of the regression line.

## Control experiment

Eleven subjects (average age: 23 years; range 20–31; 8 females, all right handed) were enrolled in a control experiment to test the influence of the object boundary on cue visibility. Stimuli and procedures were similar to Experiment 1B with the following exceptions: Part of the wedge shape around the black fixation point was removed and a circular window of 1.4° diameter around fixation was white to remove the difference between the two wedges. The participants indicated whether the cue was red or green and we computed the correlation between RT and the distance to the boundary. We did not observe a significant correlation, which indicates that the boundaries did not mask the colored cues in the main experiments.

## Experiment 2

### Participants

Eighty-nine participants (average age: 21 years; range: 18–29; 62 females, 75 right-handed) enrolled in Experiment 2. We removed one participant from the dataset because eye tracking failed when she fell asleep (the pattern of RTs was similar to other participants) so that 88 participants remained.

### Stimuli and procedure

Participants judged whether two cues were placed on the same or different objects. Trials started with the presentation of a white fixation point on a black background (*Figure 5Fig. 5a*). When the participants had maintained gaze at the fixation point for 500 ms, the second white cue (0.5°) appeared for 1000 ms so that subjects could register its location before stimulus onset. Then an image (maximal size: 27° x 20°) of two objects was presented with the fixation point and second cue superimposed. The pictures were presented in full color; the cartoon versions of the stimulus were presented as white lines on a black background (*Figure 5—figure supplement 1*). At the onset of the image, the cues started flickering at 10 Hz to ensure their continuous visibility. Participants indicated their same/different answer by a left/right button press (the assignment of buttons was counterbalanced across subjects). The stimulus disappeared after a response was given or after 5000 ms. Feedback ('correct', 'incorrect', or 'too late') was given after every trial.

For a given picture, the first cue was the fixation point, which was always at the same position for that particular image and the second cue was shown in one of six positions (within-subject factor). Three cues were on the same object (50% of trials) and three on a different object (the other 50% of trials) (*Figure 5b*). The distance between the second cue and fixation was 4.7° (1/3 of trials) or 9.4°. (2/3 of trials; 1/3 per cue with this distance). We matched the distances of the cues presented on the 'same' and 'different' object to ensure that they were not informative about the required response.

The four image conditions were a between-subject factor: color pictures, detailed cartoons, cartoon outlines, or scrambled cartoons (*Figure 6*, left panels). There were twenty-four images per condition (*Figure 5—figure supplement 1*); twelve pictures with two animals and twelve pictures with two vehicles, adapted from *Korjoukov et al., 2012*. The 88 participants were distributed equally over the four image conditions. We constructed the scrambled cartoon images by scrambling a few line segments of the cartoon outline images so that they became difficult to recognize, but we did not change the image region where the two objects intersected.

We instructed the participants to perform the task as accurately and quickly as possible. During the explanation of the task we showed all the images once and participants performed 24 practice trials before data collection started to ensure that they understood the task. The participants completed six blocks of 144 trials each, resulting in a total of 864 trials. Within blocks, we randomized the trials of the six cue-conditions and pictures. We required 64 participants (equally distributed across image conditions) to maintain fixation throughout the trial. Whenever fixation was lost (i.e., deviated from the fixation point by more than 1.5°), the trial was aborted and repeated again later within the same block. The other 24 participants directed their gaze to the fixation point to start the trial, but they were then allowed to make eye movements during the remainder of the trial. Previous studies using related tasks (*Jolicoeur and Ingleton, 1991*) and stimulus material (*Korjoukov et al., 2012*) demonstrated that eye movements do not have a strong influence on RT. Indeed, the RTs of

subjects that only fixated at the beginning of the trial were highly correlated with those of subjects who maintained fixation (r=0.83, t(70)=12.63, p<0.0001), indicating that eye movements did not have a large influence on RTs. Our main analyses were therefore based on the combined dataset from both experiments. However, if we evaluated the data of the 64 participants that maintained fixation separately, we reproduced the same conclusions: the correlation of the growth-cone model prediction with the RTs to the scrambled cartoons was 0.67, which was significantly better than all other models (p<0.01 in a bootstrap test); the correlation with the data obtained with the cartoon outlines was 0.59, which was significantly better than the other models (p<0.05 in a bootstrap test). In the data set of the 24 participants that were allowed to make saccades we found the same pattern of results: the correlation of the growth-cone model in the scrambled cartoons was 0.81, superior to all other models (p<0.01, bootstrap test); and the correlation of the model prediction with the cartoon outline data-set was 0.55, again, better than the other models (p<0.01, bootstrap test).

## Data analysis

We analyzed the pattern of RTs of correct trials after removing outliers from each of the six cue conditions that deviated from the mean of the 1/RT distribution by more than 2.5 s.d. (<2% of the data). We tested for the presence of a speed-accuracy trade-off in the four image-conditions by correlating the average RT and accuracy across cue conditions, but did not obtain evidence for such a trade-off in any condition. We used a regression analysis to test how well $RT_{Fill}$, $RT_{Eucl}$, $RT_{Pix}$, and $RT_{GC}$ (based on equations 1–4) predict the observed RT in the various cue conditions. We tested differences in the quality of the fits between models with the bootstrapping procedure that has been described for Experiment 1.

## Experiment 3

### Participants

Eleven participants (average age: 24 years; range: 20–36; 9 females, 10 right-handed) enrolled in Experiment 3.

### Stimuli and procedure

A trial (see *Figure 7a*) started when the participant fixated on a red fixation point (0.4° in diameter) on a black background for 500 ms. White outlines of two horizontal or vertical bars (broad bars: both bars 3.9° wide and 15.6° long; or narrow bars: 2.0° wide and 15.6° long) were presented with one white cue on one end of one of the bars for 100 ms (line thickness of the bars: 0.2°; line thickness of the cue: 1.2°). The cue extended over the full width of the bar and for 3.9° on the long side of the bar. After cue offset, the bars remained on the screen and participants pressed a button when they detected a red target dot (present in 80% of trials), which appeared at a cue-target onset asynchrony of 200, 300, 400, 500, or 600 ms measured relative to the offset of the cue. The red target dot (0.8° diameter) was presented at one of three locations (distance to fixation: 8.3°) until the button press. The remaining 20 percent of trials were catch trials and the participant had to maintain fixation for 1000 ms. Feedback was given at the end of every trial for correct (for 500 ms) or incorrect (for 3000 ms) responses.

A target trial could be 1) valid (60% of non-catch trials): the target was presented at the cued location; 2) invalid on the same object (20%): the cue and target are presented on opposite sides of the same bar; or 3) invalid on the different object (20%): the cue and target are presented on different bars (inset in *Figure 7a*). Note that the distance between the cue and the target was identical for the invalid same and invalid different conditions. On each trial, both bars were either broad or narrow. All trial types (3 validity types at a ratio of 3:1:1, horizontal/vertical orientation of the bars, 4 corners, 2 bar widths, 5 cue-target onset asynchronies) were presented twice, resulting in a total of 800 target trials and we interleaved 200 catch trials with a random orientation, cued corner, and bar width. The trial was aborted and repeated later if fixation was lost (gaze deviated by more than 2° from fixation) or an error was made. The participants completed 10 blocks of 100 trials. They could proceed to the next block whenever they were ready by pressing a button, but after even-numbered blocks, participants had to take a break of at least 1 min. We instructed the participants to perform the task as accurate and fast as possible.

## Data analysis

We analyzed the RTs of correct trials after removing outliers for each condition (<2.5% of the data) by excluding data points that deviated from the mean of the 1/RT distribution by more than 2.5 s.d. We used a paired t-test to assess the difference between RTs on valid trials with broad and narrow bars and we used an additional paired t-test to test the RT difference between valid and invalid trials (*Figure 7b*). We computed the object-based advantage by subtracting RTs to the invalidly cued targets on the same bar from those on the other bar, separately for broad and narrow bars and for each of the five cue-target onset asynchronies. We fitted a Gaussian function to the object-based RT advantage as function of onset asynchrony to determine the optimal cue-target onset asynchrony for the two bar widths (*Figure 7c*). We applied a bootstrapping procedure to assess the reliability of the peaks of the fitted Gaussian functions. We selected a random sample of subjects (with replacement) and fitted Gaussian functions to the time course of the object-based advantage. We repeated this procedure 10,000 times to compute the 95% confidence interval of the peak time for broad and narrow bars. We used the growth-cone model to determine the number of growth cones between the cue and invalid target position on the same object (3.3 growth-cones for the broad bar; 7.2 for the narrow bar) and computed the speed of the attentional spreading process by dividing the time-difference between the peak same object-advantages for the broad and narrow bar by the difference in the number of growth-cones.

## Acknowledgements

The authors thank Heiko Neumann for helpful comments on a earlier draft of the manuscript, Marita Rokx, Kaushik Lakshminarasimhan, Klaudia Ambroziak, and Sasa Kozelj for help with collecting the data, and Laurens van der Maaten for help with the model implementation.

# Additional information

### Funding

| Funder | Grant reference number | Author |
| --- | --- | --- |
| Nederlandse Organisatie voor Wetenschappelijk Onderzoek | VICI scheme, MaGW grant, 400-09-198 | Pieter R Roelfsema |
| Nederlandse Organisatie voor Wetenschappelijk Onderzoek | Brain and Cognition grant, 433-09-208 | Pieter R Roelfsema |
| Nederlandse Organisatie voor Wetenschappelijk Onderzoek | ALW 823-02-010 | Pieter R Roelfsema |
| European Research Council | PITN-GA-2011-290011, ABC | Pieter R Roelfsema |
| European Research Council | ERC advanced grant, 39490, Cortic_al_gorithms | Pieter R Roelfsema |
| European Research Council | 604102, Human Brain Project | Pieter R Roelfsema |

The funders had no role in study design, data collection and interpretation, or the decision to submit the work for publication.

### Author contributions

DJ, Designed the experiment, Made the stimuli, Implemented the models, Performed the experiments, Analyzed the data, Did the statistical analysis, Wrote the manuscript; MWS, Designed the experiment, Did the statistical analysis, Wrote the manuscript; PRR, Designed the experiment, Wrote the manuscript

### Author ORCIDs

Danique Jeurissen, http://orcid.org/0000-0003-3835-5977
Matthew W Self, http://orcid.org/0000-0001-5731-579X
Pieter R Roelfsema, http://orcid.org/0000-0002-1625-0034

### Ethics

Human subjects: The Ethics Committee at the University of Amsterdam approved the experiments. Informed consent was obtained from the participant before the start of the experiment.

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
