## [Decision Letter]

Thank you for submitting your article "Serial grouping of 2D-image regions with object-based attention in humans" for consideration by *eLife*. Your article has been reviewed by two peer reviewers, including Ruediger von der Heydt, and the evaluation has been overseen by Sabine Kastner as the Reviewing Editor and David Van Essen as the Senior Editor.

The reviewers have discussed the reviews with one another and the Reviewing Editor has drafted this decision to help you prepare a revised submission.

Summary:

The mechanisms underlying the process of perceptual organization of multiple object displays or scenes are poorly understood. The present paper uses behavioral data to derive a 'growth-cone' model that describes the propagation of visual signals across the processing hierarchy with the help of object-based attentional spreading.

Essential revisions:

We received two complementary reviews that were supportive and enthusiastic about your study. Three major issues deserve further consideration: (i) The non-linearity of the mapping of retinal to cortical space and the consequences for signal propagation need to be discussed (please make your assumptions regarding linearity explicit.); (ii) Details of the growth-cone model need to be explained. (iii) Difficulties of the model regarding texture and internal contours need to be addressed.

*Reviewer #1:*

This is a wonderful study of perceptual grouping in the human visual system. By measuring the reaction time of subjects' decisions whether two point probes are on the same object or not, the authors provide strong support for a "growth-cone model" compared to several other relevant models. The study also measures the time course of attention spreading with an adaptation of the Egly et al. paradigm, again in support of their model. The study and presentation of the results seems fine to me, except for one major concern that is outlined below.

My concern is that the way the authors link the model to the structure of the visual cortex assumes that retinal space is mapped linearly onto cortical space. Under this assumption, visual propagation velocity measured in °/s corresponds to neural propagation velocity measured in growth-cones/s, and cm/s in each cortical area. With this assumption, the data suggest a fixed neural propagation velocity the estimate of which is in reasonable agreement with known estimates of horizontal fiber conduction velocity. However, as is well known, cortical mapping is not linear, but follows a logarithmic law. It seems to me that, taking the real physiology into account, the model would not predict a linear relationship as shown, for example, in Figure 4 (right). While the authors take pains to discuss their somewhat lower estimate of propagation velocity compared to estimates of horizontal fiber conduction, I could not find any discussion of the mapping assumption in this paper, not even in the supplementary information. This issue needs to be addressed in Results (if appropriate by additional computations) and certainly in the Discussion.

*Reviewer #2:*

I think that this is a nice study with interesting results. I recommend acceptance if the issues below are adequately addressed.

A general issue that requires a better, more coherent treatment has to do with the roles of bounding contour, internal contours, and surface properties in the process of spreading attention. Without an adequate discussion the models are not sufficiently clear, and it will improve the discussion of the experimental results.

For example, in presenting the 'growth-cone' model (subsection “Models of perceptual grouping”), it is stated that cone 'should not touch the boundaries'. This immediately raises a difficulty in the reader's mind: do we have to know the object external boundaries in order to apply the process? This brings about a potential chicken-and-egg problem, where some sort of object segmentation is performed prior to the application of the process.

A similar difficulty arises with the notion of surface properties. For example, the model of uniform connectedness (subsection “Models of perceptual grouping”) talks about 'homogenous surface properties', but it remains unclear whether the two different shapes have the same or distinct surface properties. The two cases give naturally rise to different predictions; in the case of two uniform, but different shapes, the task would be in principle much easier.

A clear understanding of the models will therefore require a discussion of the possible effects of internal contours and surface properties. This will also be useful for discussion the experimental results of experiment 2, where the growth-cone model runs into difficulties with shapes that have internal contours, and with colored shapes.

In the discussion of experiment 2 (subsection “Influence of object recognition on image parsing”, last paragraph) the discussion concludes that 'interior contours, as well the influence of color and texture go beyond the present growth-cone model so that variance had to remain unexplained.' I think that the discussion could benefit here from some conceptual aspects. For example, explain why the growth-cone process as used in the model runs into difficulties with internal boundaries and discuss potential implications, does it mean for instance that some form of distinguishing internal from external boundaries is implied prior to the application of the attention-spread process?

In the subsection “Models of perceptual grouping”: Uniform connectedness – not sufficiently clear. I think, if the two shapes have different colors, e.g. on is red and the other is green, you can check if the two marked locations are on the same shape. But the shapes are defined here by the bounding contours, and the inside surface properties are the same.

In the subsection “Models of perceptual grouping”, Growth-cone model: it will useful to add a reference to Visual Routines (Ullman 1984) for discussion of 'attention spread' including scale-independent attention spread.

The Filling-in model: do we have to know which contours are boundaries or which side is the inside?

The wedge stimulus, experiment 1. The wedges are example of difficulty with possible internal contours. It is not entirely clear if the line separating the two wedges is an internal or external contour.

In the fourth paragraph of the subsection “Experiment 2 – Perceptual grouping of complex shapes”. 'Thus, perceptual grouping of 2D image regions invokes a serial grouping process with a speed that depends on their scale.' It is not sufficiently clear, what the 'scale' here refers to (scale of what?).

In the last paragraph of the subsection “Experiment 2 – Perceptual grouping of complex shapes”. The 'growth-cone' had problems with 'detailed' cartoons – does 'detailed' here mean the presence of internal contours? Will be useful to state this here explicitly.

In the first paragraph of the subsection “Experiment 3 – The role of object-based attention in perceptual grouping”. Give a brief description of the Egly method.

---

## [Author Response]

Essential revisions:

We received two complementary reviews that were supportive and enthusiastic about your study. Three major issues deserve further consideration: (i) The non-linearity of the mapping of retinal to cortical space and the consequences for signal propagation need to be discussed (please make your assumptions regarding linearity explicit.); (ii) Details of the growth-cone model need to be explained. (iii) Difficulties of the model regarding texture and internal contours need to be addressed.

Reviewer #1:

This is a wonderful study of perceptual grouping in the human visual system. By measuring the reaction time of subjects' decisions whether two point probes are on the same object or not, the authors provide strong support for a "growth-cone model" compared to several other relevant models. The study also measures the time course of attention spreading with an adaptation of the Egly et al. paradigm, again in support of their model. The study and presentation of the results seems fine to me, except for one major concern that is outlined below.

My concern is that the way the authors link the model to the structure of the visual cortex assumes that retinal space is mapped linearly onto cortical space. Under this assumption, visual propagation velocity measured in °/s corresponds to neural propagation velocity measured in growth-cones/s, and cm/s in each cortical area. With this assumption, the data suggest a fixed neural propagation velocity the estimate of which is in reasonable agreement with known estimates of horizontal fiber conduction velocity. However, as is well known, cortical mapping is not linear, but follows a logarithmic law. It seems to me that, taking the real physiology into account, the model would not predict a linear relationship as shown, for example, in Figure 4 (right). While the authors take pains to discuss their somewhat lower estimate of propagation velocity compared to estimates of horizontal fiber conduction, I could not find any discussion of the mapping assumption in this paper, not even in the supplementary information. This issue needs to be addressed in Results (if appropriate by additional computations) and certainly in the Discussion.

The reviewer raises an interesting point about the dependence of receptive field size on eccentricity: more eccentric RFs in a single visual cortical area are larger. What are the implications of this dependence for the growth-cone model?

It is perhaps remarkable, but the implications turn out to be minimal. The reason is that the RF sizes of the growth-cone model are not mapped onto specific visual cortical areas. If grouping demands a specific growth-cone size (as measured in °), the required RFs near the fovea will be in higher visual areas than those at more peripheral locations. Suppose the task demands grouping image patches with a size of ~2°: Depending on eccentricity, grouping can occur at a low level (e.g. V1 if eccentricity is ~10°) (Gattass et al. 1981) or at a higher level (e.g. V4 if eccentricity is ~2°) (Gattass et al. 1988). Thus, if a single elongated bar with a width of 2° needs to be grouped (as in our variant of the Egly et al. paradigm, Figure 7) the same grouping process involves V4 neurons at 2° eccentricity and V1 neurons at 10° eccentricity. In V1, neurons with abutting, non-overlapping RFs are separated by 2-4 mm (Hubel & Wiesel 1974; Harvey & Dumoulin 2011), and the distance between neurons with non-overlapping RFs (the “population point image”) is only slightly larger in higher visual areas (Harvey & Dumoulin 2011). It seems likely that the propagation of enhanced neuronal activity through horizontal connections occurs at similar speeds (in m/s) in different visual areas, and it therefore follows that the grouping speed (in °/s or in growth-cone/s) is constant across eccentricities. The growth-cone mechanism is therefore fully compatible with receptive field sizes increasing with eccentricity.

We clarified this issue in a new paragraph (subsection “A growth-cone model for parsing 2D-image regions”, last paragraph) of the Discussion.

Reviewer #2:

I think that this is a nice study with interesting results. I recommend acceptance if the issues below are adequately addressed.

A general issue that requires a better, more coherent treatment has to do with the roles of bounding contour, internal contours, and surface properties in the process of spreading attention. Without an adequate discussion the models are not sufficiently clear, and it will improve the discussion of the experimental results.

For example, in presenting the 'growth-cone' model (subsection “Models of perceptual grouping”), it is stated that cone 'should not touch the boundaries'. This immediately raises a difficulty in the reader's mind: do we have to know the object external boundaries in order to apply the process? This brings about a potential chicken-and-egg problem, where some sort of object segmentation is performed prior to the application of the process.

We thank the reviewer for pointing out this important issue. We observed that the growth-cone model is only superior to the other models for pictures without interior contours (Figure 6) and we agree that it does not solve the chicken-and-egg problem of distinguishing between internal and external contours. It only accounts for reaction time data within a more limited domain: line drawings with homogenous surfaces and bounding (external) contours.

Yet, perceptual grouping in natural images (Korjoukov et al., 2012) also relies on a serial grouping process, just as is the case for the line drawings without internal contours (Figure 4, Figure 5). Human observers are able to distinguish interior from exterior contours in the pictures of the present study (Figure 5—figure supplement 1). Consider, for example, the line drawing of the two trucks in Figure 9. We can see that the contour indicated by the arrow is an external contour of the left truck. Note, however, that this percept relies on our knowledge about the shape of trucks. Segmentation that is based on the characteristic shapes of objects is known as “semantic segmentation”.

Author response image 1.We see that the contour indicated by the red arrow is part of the left truck.The correct assignment of this contour depends on knowledge about the shape of a truck. Image parsing based on shape knowledge is called “semantic segmentation”.**DOI:**
http://dx.doi.org/10.7554/eLife.14320.012

Semantic segmentation is beyond the capabilities of all the simple models that we have considered in the present manuscript. However, in the revision we now also indicate how future incremental grouping models could take advantage of the new insights derived from convolutional neural networks for object recognition that are composed of multiple hierarchically organized layers (e.g. Krizhevsky et al. (2012); Zeiler & Fergus (2012); see also the review by LeCun et al. (2015)). These convolutional networks provide a useful description of the computations that implement object recognition in the ventral stream, from early visual cortical areas toward the inferotemporal cortex (Güçlü & van Gerven, 2015; Yamins & DiCarlo, 2016). Recent extensions of hierarchical convolutional networks have been used for semantic segmentation. They can label the image elements that belong to one of the image categories in pictures of everyday visual scenes (Hong et al., 2015; Noh et al., 2015). They solve the chicken-and-egg problem by first determining the semantic categories present in the visual scene with a convolutional network and they then use a “deconvolutional network” to go from one of the categories back to the low-level features that are part of the objects of this category. It is conceivable that these convolutional/deconvolutional networks, in combination with a horizontal grouping process, might account for the serial patterns of reaction times in grouping tasks with colorful pictures, textures and interior contours (as in Korjoukov et al., 2012). Although this is a promising avenue for future research, these more complex models remain to be built, as we now indicate in a new section of the Discussion (“Influence of object recognition and interior contours on image parsing”).

A similar difficulty arises with the notion of surface properties. For example, the model of uniform connectedness (subsection “Models of perceptual grouping”) talks about 'homogenous surface properties', but it remains unclear whether the two different shapes have the same or distinct surface properties. The two cases give naturally rise to different predictions; in the case of two uniform, but different shapes, the task would be in principle much easier.

We agree with the reviewer that grouping is easier if each shape has its own unique surface properties (e.g. a unique color). We note, however, that Palmer and Rock (1994) claimed that multiple regions are automatically parsed in separate uniform connected regions, irrespective of whether they have the same or different surface properties. This rule has been rather influential (the paper received 556 citations according to Google Scholar), which is why we used it as a baseline for our results.

In a recent study, Watson et al. (2013) investigated the effect of line properties in a curve-tracing task, presenting curves with the same or different contrast polarities. As anticipated by the reviewer, grouping was fastest if the target curve had a unique contrast polarity, a finding that provides additional evidence against the “uniform connectedness” rule. We have now more carefully described the “uniform connectedness rule”, clearly indicating that this was a previous proposal by Palmer & Rock (1994). See subsection “Models of perceptual grouping” and the first paragraph of the Discussion.

A clear understanding of the models will therefore require a discussion of the possible effects of internal contours and surface properties. This will also be useful for discussion the experimental results of experiment 2, where the growth-cone model runs into difficulties with shapes that have internal contours, and with colored shapes.

In the discussion of experiment 2 (subsection “Influence of object recognition on image parsing”, last paragraph) the discussion concludes that 'interior contours, as well the influence of color and texture go beyond the present growth-cone model so that variance had to remain unexplained.' I think that the discussion could benefit here from some conceptual aspects. For example, explain why the growth-cone process as used in the model runs into difficulties with internal boundaries and discuss potential implications, does it mean for instance that some form of distinguishing internal from external boundaries is implied prior to the application of the attention-spread process?

We hope to have addressed this issue satisfactorily in our reply to one of the reviewers’ previous issues.

In the subsection “Models of perceptual grouping”: Uniform connectedness – not sufficiently clear. I think, if the two shapes have different colors, e.g. on is red and the other is green, you can check if the two marked locations are on the same shape. But the shapes are defined here by the bounding contours, and the inside surface properties are the same.

We also hope to have addressed this point in the above because we have now clarified that the uniform connectedness rule was proposed by Palmer and Rock (1994).

In the subsection “Models of perceptual grouping”, Growth-cone model: it will useful to add a reference to Visual Routines (Ullman 1984) for discussion of 'attention spread' including scale-independent attention spread.

We have added the reference to Ullman (1984) in this section and we now indicate that previous models for curve-tracing were inspired by the visual routines hypothesis.

The Filling-in model: do we have to know which contours are boundaries or which side is the inside?

The wedge stimulus, experiment 1. The wedges are example of difficulty with possible internal contours. It is not entirely clear if the line separating the two wedges is an internal or external contour.

The simple models in the manuscript assume that all contours are external and they are unable to segment images with internal contours. Thus, these models form the correct groupings in the wedge experiment by simply assuming that the line separating the wedges is external. Furthermore, in the experiment participants were instructed to select individual wedges as separate objects.

We do appreciate the reviewer’s concern because most objects in realistic images do contain internal contours. In the revised Discussion we explain that parsing of these more realistic images will require feedback from shape representations. This process is known as “semantic segmentation” as has been outlined in our response to one of the earlier remarks of the reviewer.

In the fourth paragraph of the subsection “Experiment 2 – Perceptual grouping of complex shapes”. 'Thus, perceptual grouping of 2D image regions invokes a serial grouping process with a speed that depends on their scale.' It is not sufficiently clear, what the 'scale' here refers to (scale of what?).

We have clarified what we mean by 'scale': grouping is faster for wide than for narrow image regions (subsection “Experiment 2 – Perceptual grouping of complex shapes”, fourth paragraph).

In the last paragraph of the subsection “Experiment 2 – Perceptual grouping of complex shapes”. The 'growth-cone' had problems with 'detailed' cartoons – does 'detailed' here mean the presence of internal contours? Will be useful to state this here explicitly.

Yes, the 'detailed cartoons' refers to the image set with internal contours. We now explicitly state this in the fifth paragraph of the subsection “Experiment 2 – Perceptual grouping of complex shapes”.

In the first paragraph of the subsection “Experiment 3 – The role of object-based attention in perceptual grouping”. Give a brief description of the Egly method.

We added a brief description of the paradigm and main findings of Egly et al. (1994) on page 12.